## Overview Review

mental health policy; Nigeria; policy implementation; health systems strengthening; scoping review; Walt and Gilson's policy triangle; equity and access; low- and middle-income countries

**Corresponding author:**
Benmun Damul;
Email: benmundamul@gmail.com

# Mental health policy interventions in Nigeria: A scoping review of development, implementation and outcomes

Benmun Damul[1] [iD], Hazel Carolyn King[2], Halima Jafiya[2] and Christine Bestman[3]

[1]Public Health, Vanderbilt University, USA; [2]Mental Health Research and Policy Hub, Nigeria and [3]The Leprosy Mission, Nigeria

## Abstract

Approximately 20% of Nigerians experience a mental health condition, yet fewer than 10% receive minimally adequate care. This scoping review synthesises the development, implementation and outcomes of mental health policies in Nigeria from 1916 to 2025. Using Arksey and O'Malley's framework, systematic searches were conducted across PubMed, Web of Science, PsycINFO, AJOL and Google Scholar (inception–December 2024), supplemented by grey literature from governmental and non-governmental sources. The Walt and Gilson Policy Triangle guided the analysis of policy context, content, processes and actors. Nigeria's policy trajectory demonstrates normative progress, transitioning from custodial approaches under the Lunacy Ordinance (1916) to a rights-based orientation in the Mental Health Act (2023). However, implementation outcomes remain constrained. Workforce expansion has been modest (psychiatrists increased from 250 in 2018 to approximately 350 in 2024), treatment coverage remains low (10–15%) and budget allocation is insufficient (3.3% of the health budget). Barriers include inadequate financing, weak coordination across federal and state levels, limited stakeholder engagement and insufficient integration of community, traditional and faith-based providers. Comparative analysis highlights that Ghana's autonomous Mental Health Authority, South Africa's provincial directorates and Kenya's community health volunteer model provide governance and implementation structures absent in Nigeria. Findings indicate that Nigerian mental health policies, while necessary, are insufficient alone for system strengthening. Effective policy translation requires increased and protected financing (target: 5% of health budget by 2027), task-shifting strategies, establishment of a National Mental Health Information System, federal incentives for state-level adoption, integration into primary healthcare (5,000 PHCs by 2028) and inclusive governance that incorporates service users and traditional healers.

## Impact Statements

Mental health policy reforms in Nigeria exemplify a broader low- and middle-income country challenge: translating progressive legislation into equitable, scalable service delivery within resource-constrained, decentralised systems. This scoping review maps policy evolution from the custodial Lunacy Ordinance (1916) to the rights-based Mental Health Act (2023), analysing implementation using the Walt and Gilson Policy Triangle framework (context, content, processes and actors). Findings reveal persistent policy–practice gaps. Although Nigeria has enacted comprehensive mental health legislation, implementation remains constrained by chronic underinvestment (3.3% of health budget), severe specialist shortages (0.15 psychiatrists per 100,000 population) and fragmented federal–state–local governance. Legislative adoption lacks commensurate budgetary commitments, operational frameworks or monitoring mechanisms. Consequently, policy intent has not translated into substantial population-level mental health improvements. Findings offer broader relevance for global mental health policy strengthening. First, the Policy Triangle elucidates barriers spanning contextual factors (fiscal austerity and political instability), content limitations (insufficient costing and operational clarity), process weaknesses (limited stakeholder engagement) and actor dynamics (restricted participation of service users, traditional healers and community actors). Second, Ghana, South Africa and Kenya demonstrate that autonomous mental health authorities with protected funding outperform units embedded within general health ministries. Third, uniform policy approaches address marginalised subpopulations inadequately: women, internally displaced persons, rural residents, adolescents and individuals with severe mental illness. By synthesising implementation bottlenecks and highlighting feasible, scalable strategies – primary healthcare integration and task-shifting – the review provides evidence-informed guidance for policymakers, practitioners and global health partners. The analysis advances scholarship on mental health systems in federal, pluralistic contexts and supports achieving Sustainable Development Goal 3.4.

## Introduction

Mental and behavioural disorders account for approximately 7.3% of the global disease burden, with depressive disorders, anxiety, psychoses and substance use disorders representing predominant conditions (WHO, 2013; GBD, 2017). Globally, 450 million people experience mental disorders, with one in four individuals developing a mental disorder during their lifetime (Uddin et al., 2019). This burden disproportionately affects low- and middle-income countries, where 75% of people with mental disorders receive no treatment – the "treatment gap" (Patel et al., 2018).

Nigeria, with a population of approximately 223 million (2024 estimate), faces substantial mental health challenges. Approximately 20% of the population experiences mental health conditions (Mbamalu, 2019; API, 2020), yet fewer than 10% access appropriate care (Fadele et al., 2024). The mental health workforce remains critically insufficient: Nigeria has approximately 350 psychiatrists (0.16 per 100,000 population), far below WHO recommendations (1.0 per 100,000) and regional comparators such as South Africa (0.28 per 100,000) (Agbo, 2022; WHO, 2024).

Beyond workforce limitations, pervasive stigma constrains care access. Mental illness is frequently attributed to supernatural causes – divine punishment, witchcraft and ancestral curses – leading to social exclusion and preference for traditional or faith-based healers (Labinjo et al., 2020; Lawal et al., 2024). Structural barriers – geographic inaccessibility, prohibitive costs and limited service availability – intersect with stigma to exacerbate the treatment gap (Ikwuka et al., 2016).

Despite these challenges, Nigeria's policy environment has evolved substantially. From the colonial era Lunacy Ordinance (1916) to the Mental Health Act (2023), policies have progressively embraced rights-based, community-oriented approaches. Yet implementation deficits persist: policies remain inadequately resourced, poorly coordinated across federal–state–local levels and insufficiently integrated with community realities. Understanding why well-intentioned policies fail is critical for Nigeria and other LMICs facing similar constraints.

### Study rationale and objectives

This scoping review addresses a critical gap: while individual policies have been described, no comprehensive analysis has examined the policy landscape through an implementation lens. Existing literature is largely descriptive; this review provides critical, theoretically grounded analysis.

**Objectives:**

1. Map Nigeria's mental health policy landscape from the colonial era to 2025.
2. Analyse policies using the Walt and Gilson Policy Triangle framework.
3. Assess implementation outcomes and identify barriers.
4. Compare Nigeria's experience with regional examples (Ghana, South Africa and Kenya).
5. Examine differential policy impacts across subpopulations.
6. Develop evidence-informed recommendations for closing the policy–practice gap.

This review contributes to global health scholarship on policy implementation in federal, resource-constrained contexts and provides actionable guidance for policymakers, healthcare providers and mental health advocates.

## Methods

### Review design

A scoping review was conducted following the framework of Arksey and O'Malley (2005), refined by Levac et al. (2010). Scoping reviews are appropriate for mapping broad policy landscapes, identifying key interventions and synthesising diverse evidence types – including legislation, policy documents, implementation studies and grey literature – particularly in emerging research areas where systematic reviews may be premature (Munn et al., 2018).

### Search strategy

#### Electronic databases

Systematic searches were conducted from database inception to December 2024 across PubMed/MEDLINE, Web of Science, PsycINFO, African Journals Online (AJOL) and Google Scholar.

#### Search terms

Three concept blocks were combined using Boolean operators:

1. **Mental health terms:** "mental health" OR "mental disorder" OR "psychiatric" OR "psychosocial" OR "mental illness"
2. **Policy terms:** "policy" OR "policies" OR "legislation" OR "law" OR "act" OR "health system" OR "reform" OR "intervention" OR "service delivery"
3. **Geographic terms:** "Nigeria" OR "Nigerian"

#### Grey literature

Grey literature sources: Federal Ministry of Health, National Primary Healthcare Development Agency, WHO Nigeria, UNICEF Nigeria, BasicNeeds, CBM Global, MANI, Mental Health Foundation Nigeria, National Assembly records and citation chaining.

### Inclusion and exclusion criteria

**Inclusion criteria:**

1. Documents addressing mental health policy, legislation or system-level interventions in Nigeria
2. English language publications
3. Peer-reviewed or authenticated grey literature
4. Empirical studies, policy analyses or authoritative commentary

**Exclusion criteria:**

1. Clinical intervention studies without policy implications
2. Opinion pieces lacking empirical or authoritative grounding
3. Documents inaccessible in full text
4. Non-English publications (no relevant vernacular literature identified)

### Study selection process

Two independent reviewers screened titles and abstracts against predefined inclusion criteria. Agreement between reviewers was excellent (Cohen's $\kappa = 0.89$). Full texts of potentially eligible documents were assessed for inclusion, with discrepancies resolved through discussion or adjudication by a third reviewer. Reference lists of included documents were hand-searched, yielding 12 additional sources.

### Data extraction

A standardised extraction form was captured:

1. **Policy characteristics:** title, year, type (legislation, policy, strategic framework)
2. **Key provisions:** scope, target populations, service delivery models, governance structures
3. **Implementation mechanisms:** funding, responsible agencies, monitoring frameworks
4. **Reported outcomes:** coverage data, workforce statistics, budget allocations
5. **Identified barriers:** resource constraints, coordination challenges, stakeholder engagement gaps

One reviewer extracted data, and a second verified accuracy. Discrepancies were resolved by consensus.

### Analytical framework

The Walt and Gilson Policy Triangle (Walt and Gilson, 1994) guided the analysis across four dimensions: context (historical, social, economic and political factors); content (specific provisions, scope and design); processes (policy development, adoption, implementation pathways and stakeholder engagement) and actors (key stakeholders, power dynamics and institutional roles). The framework is suited for analysing health policy in LMICs and federal systems (Gilson and Raphaely, 2008; Buse et al., 2012).

### Synthesis approach

Findings were synthesised narratively and organised:

- **Chronologically:** Colonial era (1906–1960), post-independence (1960–1999), democratic era (1999–2013), contemporary period (2013–2025)
- **Thematically:** Policy evolution, implementation landscape, comparative insights, subpopulation considerations
- **Through Policy Triangle dimensions:** Systematically analysing each major policy intervention using the four-dimensional framework

### Methodological limitations

Limitations include publication bias in government grey literature, paucity of routine outcome data, language restriction to English sources and incomplete capture of recent developments (2024–2025). Despite these limitations, this review provides the most comprehensive, theoretically grounded synthesis of Nigerian mental health policy to date.

## Findings

### Colonial era to Independence (1906–1960): Custodial beginnings

#### Context

Prior to colonial influence, mental healthcare in Nigeria was embedded in traditional frameworks. Mental illness was attributed to spiritual imbalances, ancestral displeasure or witchcraft, with traditional healers providing treatment through rituals, herbal remedies and spiritual interventions (Teferra and Shibre, 2012; Steinforth, 2021). British colonisation introduced Western biomedical psychiatry, fundamentally reshaping responses to mental illness.

#### Content

The first formal mental health facilities were established under colonial administration: Calabar Asylum (1904) and Yaba Asylum (1907) in Lagos (Sadowsky, 1999). These institutions reflected a custodial approach prioritising confinement over rehabilitation. The Lunacy Ordinance (1916) codified this approach, labelling individuals as "lunatics" or "idiots" and granting magistrates authority to detain indefinitely, with no definitions regarding severity or treatment need (Kingdon, 1959; Ugochukwu, 2020; Akanni and Edozien, 2023).

#### Processes and actors

Colonial mental health policy development systematically excluded Nigerian voices. British administrators designed mental health systems primarily as mechanisms of social control, marginalising traditional healers. Policy formation lacked therapeutic intent and prioritised population management over individual well-being.

#### Analysis

The colonial period created enduring legacies: facility-centric care, stigmatising terminology and policies divorced from cultural context. These patterns persist in contemporary Nigeria, where institutionalisation dominates, stigma is rooted in colonial era labels and integration of traditional healers remains contentious. Although amended in 1958, the Lunacy Ordinance's custodial orientation persisted until replaced by the 2023 Mental Health Act (Westbrook, 2011; Ude, 2015).

### Post-Independence to democratic era (1960–1999): Slow evolution

#### Context

Nigeria's independence in 1960 provided political space for policy reform, yet mental health remained peripheral. The 1970s oil boom increased national resources, but investment in mental healthcare was minimal. Military rule (1966–1979; 1983–1999) constrained civil society advocacy. Nonetheless, psychiatric services expanded gradually, with additional hospitals, psychiatric units in general hospitals and the emergence of professional associations.

#### Content and processes

The 1988 National Health Policy (revised 2004) acknowledged mental health but lacked actionable detail. In 1991, mental health integration into primary healthcare (PHC) represented a significant policy shift towards community-based care, aligning with the Alma-Ata Declaration (1978) (Federal Ministry of Health, 2013). PHC workers were designated first-line managers for common mental disorders. Implementation remained limited. PHC staff received insufficient training, supervision was inadequate, psychotropic medicines were scarce and political commitment was weak (Saraceno et al., 2007; Chu et al., 2022).

#### Actors

Nigerian psychiatric professionals, especially the Association of Psychiatrists in Nigeria (founded in 1969), emerged as policy actors. Broader stakeholder engagement remained minimal: service users, families and state governments were excluded, and civil society advocacy was nascent.

#### Analysis

This era exemplifies "policy without infrastructure": progressive intentions were not matched with operational capacity. No budget, training curricula, supervision structures or accountability mechanisms accompanied the PHC integration policy. This reflects

aspirational health policies detached from implementation realities (Onoka et al., 2015).

### Democratic era policy development (1999–2013): Legislative momentum

#### Context

Democracy returned in 1999, enabling civil society advocacy and legislative reform. Globally, the WHO's Mental Health Gap Action Programme (mhGAP, 2008) and recognition of mental health's role in development (later codified in SDG 3.4) heightened attention to mental health systems' strengthening. In Nigeria, psychiatric professionals, patient rights groups and international partners generated momentum for reform.

#### Content: The 2013 National Policy for Mental Health Services Delivery

The 2013 policy represented Nigeria's most comprehensive mental health framework at the time (Federal Ministry of Health, 2013). Key provisions included

1. **Primary Healthcare:** Mandatory mental health services at PHCs; essential psychotropic medicines; training for medical officers and allied health workers; strengthened community outreach and referral systems.
2. **Secondary Healthcare:** Inpatient/outpatient services at general hospitals; intersectoral governance for mental, neurological and substance abuse services.
3. **Tertiary Healthcare:** Specialised services for children, the elderly and substance use disorders; supervision for secondary facilities.
4. **Systemic Provisions:** Integration across education, social welfare and criminal justice; public–private partnerships; continuing professional education.

#### Processes: Policy triangle analysis

1. **Context:** Emerged within broader health sector reform and mhGAP influence; fiscal constraints and political instability limited implementation.
2. **Content:** Comprehensive on paper but lacked budget allocations, phased rollout and accountability mechanisms.
3. **Process Failures:** Led by the Federal Ministry of Health with WHO support; state governments, service users and traditional healers were excluded. Federal adoption did not translate to state-level action.
4. **Actors:** Federal Ministry of Health, WHO, Association of Psychiatrists; state health ministries, service users and traditional healers were absent.

**Outcome:** The policy failed to achieve objectives: PHC integration was minimal, essential medicines supply chains were absent, training was unsystematic, and no monitoring system existed. This illustrates an "implementation deficit" (Gilson and Raphaely, 2008), where policies falter due to a lack of ownership, resources and accountability.

### Contemporary policy reform (2013–2025): Legislation and implementation challenges

#### Legislative journey: The mental health bill

Efforts to replace the outdated Lunacy Act began in 2003 but were withdrawn in 2009 (Ugochukwu et al., 2020). Re-introduced in 2013, the bill sought to protect the rights of persons with mental disorders, ensure equitable access to care, discourage stigma, establish practice standards and create judicial oversight for involuntary admissions. The bill stalled until 2019, when Senator Ibrahim Oloriegbe sponsored the Mental Health and Substance Abuse Bill, signed into law as the Mental Health Act 2023, following public hearings (Akanni and Edozien, 2023; Saied, 2023).

#### Content: The 2023 mental health act

The act represents a paradigm shift from custodial to rights-based care, with provisions in three domains:

**Rights Protections:**

1. Fundamental rights of persons with mental health conditions during treatment
2. Prohibition of discrimination in employment, education and housing
3. Legal safeguards for involuntary admission with judicial oversight
4. Right to refuse treatment except in specified emergency circumstances

**Service Delivery:**

1. Mandated mental health services across all healthcare levels
2. Community-based approach emphasising recovery, rehabilitation and social integration
3. Integration into the National Health Insurance Scheme (expanding financial protection)
4. Establishment of mental health desks in State Ministries of Health

**Governance and Accountability:**

1. Creation of a Mental Health Commission (not operational as of December 2025)
2. Accreditation standards for facilities and professionals
3. Monitoring and reporting requirements
4. Penalties for rights violations

#### Complementary frameworks

The 2023 National Suicide Prevention Strategic Framework addresses rising youth suicide, emphasising risk assessment training, public awareness campaigns and strengthening services for at-risk populations (Ugoeze-Onyedika, 2023).

#### Implementation landscape (2023–2025)

Progress has been limited. Achievements: legal replacement of the Lunacy Act; federal stakeholder consultations; initial state-level policy development in Lagos and Kaduna; increased civil society advocacy. Gaps: Mental Health Commission unestablished; no operational budget; minimal state adoption (3/36 states); nonfunctional mental health desks; absent monitoring frameworks.

#### Analysis: Why has implementation lagged?

Using the Policy Triangle:

1. **Context:** The act was signed during severe fiscal constraints (2023 fuel subsidy removal, currency devaluation and high inflation). Mental health competes with more visible priorities. Political transitions (new administration, May 2023) created bureaucratic continuity challenges.

2. **Content:** Implementation is constrained by the absence of costing. No budget was attached to the act, and subsequent appropriations have not allocated funds. The act mandates services without specifying funding sources – a critical design flaw.
3. **Processes:** No robust implementation planning process was established post-enactment. The Federal Ministry of Health has not developed a phased rollout plan or state engagement mechanism. In contrast, Ghana's Mental Health Act (2012) had an implementation roadmap with timelines developed immediately post-enactment.
4. **Actors:** Key implementation actors remain under-resourced and disempowered. State governments, which must operationalise the act, lack technical and financial capacity. The proposed Mental Health Commission does not exist. Service users and families remain excluded from implementation planning.

The 2023 Mental Health Act represents necessary but insufficient progress. It establishes the legal and normative framework, but without resources, implementation infrastructure and multi-stakeholder ownership, it risks becoming another aspirational policy that fails to translate into improved services – repeating the pattern of the 2013 policy.

### Implementation outcomes: Service delivery landscape

#### Workforce: Modest expansion, persistent inadequacy

Nigeria's mental health workforce has expanded modestly but remains critically insufficient:

While numbers have increased (psychiatrists +250% since 2003), ratios remain far below WHO recommendations and regional comparators (South Africa: 0.28; Ghana: 0.19 per 100,000). Geographic concentration exacerbates inadequacy: 70% of psychiatrists practice in Lagos and Abuja, leaving northern and rural areas severely underserved (Gureje et al., 2015) see Table 1.

#### Treatment coverage: Marginal improvement

Treatment coverage improved modestly: <5% (2010) to 8–10% (2018) to 10–15% (2024 estimated), leaving 85–90% of Nigerians with mental disorders without professional care – a treatment gap consistent with other sub-Saharan African countries but unacceptable from human rights and public health perspectives (Okpalauwaekwe et al., 2017; Kola et al., 2021).

#### Service infrastructure: Urban concentration

Nigeria's mental health infrastructure (2024) includes six federal neuropsychiatric hospitals, 11 psychiatric units in teaching hospitals, approximately 15 state facilities and minimal primary healthcare integration (<5% of PHCs offer mental health services). The WHO-AIMS 2006 survey reported 1,092 psychiatric beds (3.99 per 100,000), with 86% in mental hospitals (WHO & Ministry of Health, 2006); more recent data are unavailable. The average length of stay was 52 days; 93% of admissions were under 1 year, but 1% exceeded

10 years, raising human rights concerns about prolonged institutionalisation. Essential psychotropic medicines (antipsychotic, antidepressant, mood stabiliser, anxiolytic and antiepileptic) are available at tertiary facilities but are inconsistent at the primary level, with fragile supply chains for newer medications.

#### Budget allocation: Chronic underfunding

Mental health budget allocation has remained grossly inadequate: 0.5% (2006) to 3.3% (2024) of the health budget – below international benchmarks of 5–10% (WHO, 2021). In absolute terms, Nigeria spends approximately $1.50 per capita annually on mental health compared to $30–50 in South Africa (Docrat et al., 2019; Freeman, 2022). Most funding supports the six federal neuropsychiatric hospitals, leaving minimal resources for PHC integration, community services or state-level programming.

#### Medication access: Improved but inequitable

Medication access improved incrementally: 10.4% of Nigerians with severe mental disorders accessed treatment in 2021 (Soroye et al., 2021), although access remains constrained by cost (out-of-pocket payments create catastrophic expenditures for poor families), availability (many PHCs and secondary facilities lack essential psychotropics) and knowledge (prescribers lack psychopharmacology training).

The National Health Insurance Authority's mental health benefit package (2024) covers basic psychotropic medicines but excludes newer agents and non-pharmacological interventions. Private insurance provides minimal mental health coverage, typically limited to psychiatrist consultations without psychologist services, intensive therapy or residential treatment (Coker, 2023).

#### Community-based interventions: Promising but nascent

Community-based mental health interventions have shown promise in pilot sites. The Comprehensive Community Mental Health Programme in Benue State (CBM Global) demonstrated the feasibility of integrating mental health into PHC through task-shifting: CHEWs trained in WHO mhGAP protocols successfully provided psychosocial interventions for common mental disorders, with outcomes comparable to specialist care (Abdulmalik et al., 2019; Ryan et al., 2020). Scale-up remains limited by funding constraints and a lack of political prioritisation.

NGO-led initiatives (BasicNeeds, MANI) demonstrate that community engagement, peer support integration and collaboration with traditional healers can reduce stigma and improve help-seeking (MANI, 2024). However, NGO services reach a small fraction of those in need and are donor dependent, raising sustainability concerns.

#### Digital health: Emerging frontier

Digital mental health interventions represent an emerging opportunity. Nigeria's mobile phone penetration exceeds 85%, creating potential for mHealth applications, telemedicine and digital

**Table 1.** Nigeria's mental health workforce: Growth and gaps relative to WHO minimums (2003–2024)

| Cadre | 2003 | 2018 | 2024 | Ratio per 100,000 (2024) | WHO minimum |
|---|---|---|---|---|---|
| Psychiatrists | <100 | 250 | 350 | 0.16 | 1.0 |
| Psychiatric nurses | ~500 (est.) | ~700 (est.) | ~1,000 (est.) | 0.45 | 2.0 |
| Clinical psychologists | <100 | ~300 | ~320 | 0.14 | 0.5 |

*Sources*: Omojide and Morakinyo (2003), Association of Psychiatrists in Nigeria (2018), Olatunji (2020), Agbo (2022), WHO Global Health Workforce Statistics (2024).

psychotherapy platforms (Onu and Onyeka, 2024). Pilot projects using mobile apps for medication adherence reminders, psychoeducation and symptom monitoring have shown acceptability (Wada et al., 2021). However, barriers include limited internet connectivity in rural areas (broadband penetration <40%), data costs creating accessibility barriers for low-income populations, low digital literacy and the absence of regulatory frameworks for digital mental health services.

The Federal Ministry of Health has established a Digital Health Committee but has not yet developed guidelines for telepsychiatry or digital therapeutics, creating uncertainty for providers and innovators.

### Stigma reduction: Progress among youth and urban populations

Public awareness campaigns, particularly those leveraging social media and celebrity advocates, have contributed to gradual stigma reduction, especially among younger, educated, urban populations (Abdulmalik et al., 2018). However, deeply rooted beliefs about supernatural causation persist, particularly in rural and northern regions (Okpalauwaekwe et al., 2017). Older generations remain largely resistant to biomedical mental health concepts.

Traditional and faith-based explanatory models continue to dominate: mental illness as spiritual attack, ancestral punishment or possession requiring spiritual remedies. Some faith healers incorporate biomedical concepts (e.g., praying while advising psychiatric consultation), but others actively discourage medical treatment, leading to tragic outcomes, including chaining, abuse and preventable deaths (Onyemelukwe, 2016).

### Barriers to implementation: Multi-level analysis

#### Systemic barriers

Systemic barriers include chronic underfunding (3.3% budget allocation insufficient to operationalise policy mandates), federal–state coordination gaps (healthcare delivery vests in states while policy emanates federally; states lack technical capacity, financial resources and political will), absence of monitoring systems (no national mental health information system exists; routine data on service utilisation, treatment outcomes, workforce distribution and budget execution are not collected systematically, precluding evidence-based decision-making) (Chisholm et al., 2019) and workforce development challenges (training institutions produce insufficient numbers; 30–40% of Nigerian-trained psychiatrists emigrate; task-shifting to non-specialist providers has not been systematically implemented due to lack of training infrastructure, supervision challenges and professional resistance) (Saraceno et al., 2007; Patel et al., 2018; Fadele et al., 2024).

#### Cultural and social barriers

Cultural barriers include pervasive stigma (beliefs that mental illness results from moral failure or witchcraft lead to social exclusion, delayed help-seeking with median untreated illness duration exceeding 2 years, preference for traditional healers over biomedical services and family concealment; women face compounded stigma) (Lawal et al., 2024) and limited mental health literacy (many Nigerians cannot identify common mental disorders, are unaware of effective treatments and do not know where to seek help; health education campaigns have reached urban populations, but rural communities remain underserved) (Ajike et al., 2022).

#### Structural and governance barriers

Structural barriers include weak intersectoral coordination (mental health intersects with education, social services, justice and employment, yet sectoral silos impede coordination; no functional intersectoral coordinating mechanism exists), exclusion of key actors from policy processes (service users and families whose lived experience should inform policy, traditional and faith healers who provide majority of care, community leaders who shape health-seeking behaviour and private sector providers who supplement public services; this exclusion undermines policy legitimacy and implementation feasibility) and private sector underdevelopment (the private mental health sector remains small and predominantly urban; private facilities charge fees unaffordable to most Nigerians – average outpatient consultation ₦10,000–30,000 / $12–36 USD, representing multiple days' wages; private insurance coverage for mental health is minimal, and the National Health Insurance Scheme's mental health benefit package is nascent and incompletely implemented).

## Discussion

### Policy–practice gaps

This study finds persistent implementation deficits in Nigeria's mental health system: despite progressive policies, service delivery remains minimal. The Walt and Gilson Policy Triangle framework (Walt and Gilson, 1994) illuminates mechanisms of failure across context, content, process and actors.

### Context: Structural and economic constraints

Nigeria's federal structure, fiscal constraints and political instability create significant barriers:

1. **Federal Structure:** Healthcare delivery vests primarily in states whose capacity and commitment vary widely. Wealthy states (Lagos, Rivers) face competing priorities; poorer states (Yobe, Zamfara) lack resources entirely. Federal policies that ignore state heterogeneity are unlikely to succeed (Onoka et al., 2015).
2. **Fiscal Constraints:** Nigeria's fiscal crisis – driven by oil revenue volatility, debt servicing and corruption – creates zero-sum competition for resources. Mental health, lacking political champions, receives minimal funding (₦47 billion in 2024; ~ $60 million for a population of 223 million).
3. **Political Instability:** Frequent transitions disrupt policy continuity. The 2023 Mental Health Act, signed in the final weeks of the Buhari administration, has lacked consistent implementation leadership.
4. **Global Mental Health Discourse:** While Nigeria is party to international frameworks (SDG 3.4, WHO Comprehensive Mental Health Action Plan 2013–2030), domestic compliance mechanisms are weak.

### Content: Design flaws in policy documents

Policy content is aspirational rather than actionable:

1. **Absence of Costing:** Neither the 2013 policy nor the 2023 Act included cost estimates or budget proposals. Policies without specified resource requirements are aspirational, not actionable.
2. **Lack of Implementation Timelines:** Policies provide objectives but no phased rollout plans, milestone targets or sequencing strategies, making accountability impossible.

3. **Inadequate Operational Detail:** Policies prescribe outcomes (e.g., "integrate mental health into PHC") but not operational mechanisms (training curricula, supervision and medicine supply).
4. **One-Size-Fits-None Approach:** Policies assume homogeneity across Nigeria's diverse contexts (urban/rural, north/south and wealth disparities), undermining feasibility.

*Processes: Weak stakeholder engagement and ownership*

Top-down policy development and limited consultation reduce ownership:

1. **Top-Down Approaches:** Federal government and international partners design policies with minimal state-level input. States, which must implement, lack ownership.
2. **Exclusion of Service Users:** Persons with lived experience and their families – whose perspectives are essential – have been systematically excluded from policy development, violating the "nothing about us without us" principle and reducing policy relevance.
3. **Marginalisation of Traditional Healers:** Traditional and faith healers provide the majority of mental healthcare in Nigeria, yet they are absent from policy discussions or actively opposed. Policies that ignore or antagonise dominant care providers will fail. Successful models elsewhere (e.g., Senegal, Zimbabwe) demonstrate that structured collaboration is possible and beneficial (Ae-Ngibise et al., 2010).
4. **Weak Monitoring and Accountability:** No routine mechanisms track implementation. Policies are announced, but progress is not monitored, failures are not investigated, and course corrections do not occur.

*Actors: Power dynamics and excluded voices*

Policy influence is skewed:

1. **Over-Represented:** Federal Ministry of Health, psychiatrists' associations and international organisations.
2. **Under-Represented:** State governments, service users, non-psychiatrist professionals, traditional/faith healers and private providers.

Power asymmetries ensure policies reflect the interests of over-represented actors. Psychiatrists advocate for hospital-based specialist care; community-based task-shifting models receive less support despite superior cost-effectiveness and reach.

### Comparative insights: Learning from regional mental health policy

Other African countries demonstrate effective implementation models:

a. **Ghana: Independent mental health authority**

Ghana passed the Mental Health Act 2012, 1 year before Nigeria's 2013 policy. Like Nigeria, Ghana aimed to decentralise services, protect rights and integrate into PHC (Roberts et al., 2013). However, key implementation differences emerged:

- **Implementation Architecture:** Ghana established an independent Mental Health Authority with a statutory mandate, dedicated budget line (initially 1.4% of health budget, increased to 2.1% by 2020), regulatory authority and cabinet-level leadership. Nigeria's 2023 Act proposes a Mental Health Commission,

unestablished as of December 2025, causing implementation drift.
- **Stakeholder Engagement:** Ghana's policy process involved extensive consultation with service users through the Mental Health Society of Ghana and Basic Needs Ghana, generating ownership and relevance.
- **Outcomes:** Ghana achieved broader coverage (18% of districts with services versus Nigeria's 10%) and lower documented rights violations (Ofori-Atta et al., 2018). However, Ghana still faces workforce shortages (0.19 psychiatrists per 100,000) and urban–rural inequities, demonstrating that while governance structures facilitate implementation, resource constraints remain binding.
- **Lesson for Nigeria:** Independent, empowered implementation bodies with dedicated funding are critical for translating policy into practice.

b. **South Africa: Advanced policy, persistent inequities**

South Africa's Mental Health Care Act 2002 represents the region's most advanced rights-based legislation, with comprehensive rights protections, judicial oversight for involuntary admissions, integration into the public health system at all levels, provincial mental health directorates ensuring vertical coordination and a functional mental health information system enabling monitoring.

South Africa allocates approximately 5% of its health budget to mental health – higher than Nigeria's 3.3% but still inadequate (Freeman, 2022; Shisana et al., 2013). Psychiatrist density (0.28 per 100,000) is nearly double Nigeria's, with more geographically distributed services.

- **Persistent Challenges:** Despite policy sophistication, South Africa faces profound urban–rural inequities (70% of psychiatrists in urban centres), racial disparities in access (legacy of apartheid spatial segregation), workforce retention challenges (brain drain to private sector and abroad) and implementation gaps in under-resourced provinces (Limpopo, Eastern Cape). Policy comprehensiveness did not eliminate challenges in provinces lacking resources and political will (Vergunst et al., 2018).
- **Lesson for Nigeria:** Advanced legislation, while necessary, is insufficient without addressing deep-rooted structural inequities, decentralisation challenges in federal/provincial systems and workforce distribution. South Africa's mental health information system offers a concrete model for Nigeria's proposed National Mental Health Information System (NMHIS).

c. **Kenya: Community-based innovation**

Kenya's Mental Health Policy 2015–2030 emphasises community mental health units and integration with the national Community Health Strategy (Marangu et al., 2014; Meyer and Ndetei, 2016). Kenya has prioritised task-shifting more aggressively than Nigeria:

- **Community Health Volunteers (CHVs):** Kenya trained over 50,000 CHVs in mental health first aid and identification of common mental disorders. CHVs provide psychoeducation, facilitate referrals and offer psychosocial support (Kaigwa et al., 2022).
- **Telepsychiatry:** Kenya's 2019 Mental Health Task Force prioritised telepsychiatry to overcome geographic barriers. Pilot programmes demonstrated feasibility and acceptability (Njenga et al., 2022).

- **Universal Health Coverage:** Mental health was included in Kenya's UHC pilot from inception, ensuring financial protection for common mental disorders.
- **Faith-Based Partnerships:** Kenya's Ministry of Health formalised partnerships with faith-based organisations, training religious leaders and establishing referral pathways between spiritual care and biomedical services.
- **Lesson for Nigeria:** Kenya's CHV model aligns with Nigeria's CHIPS programme. Adapting Kenya's approach could accelerate PHC integration. Integration of mental health into Nigeria's ongoing UHC expansion offers policy synergy. Formalising partnerships with faith-based organisations, rather than ignoring or opposing them, could leverage existing community structures.

*Cross-cutting lessons*

Three determinants of policy-to-practice translation emerge

1. **Implementation Architecture:** Dedicated mental health authorities or strong provincial/state structures achieve better implementation than general health ministry structures. Ghana's Mental Health Authority and South Africa's provincial directorates provide concrete models.
2. **Resource Commitment:** No country achieves policy goals without sustained budget increases. Ghana and South Africa's higher (though inadequate) allocations correlate with better coverage. Costed implementation plans are essential.
3. **Community Engagement:** Kenya and Ghana's community-based approaches may be more culturally appropriate and feasible than facility-centric models in workforce-scarce contexts. Engaging traditional and faith healers, rather than marginalising them, enhances reach and acceptability.

### Differential policy impact across subpopulations

Nigerian mental health policy largely fails to address population-specific vulnerabilities:

#### a. Women and maternal mental health

Perinatal mental disorders affect 14–23% of Nigerian women (Chinawa et al., 2015), yet maternal mental health receives minimal policy attention. The 2013 policy mentions "women" generically; the 2023 Act contains no specific provisions. This neglect occurs despite evidence that unaddressed perinatal depression contributes to maternal mortality (Nigeria's maternal mortality ratio: 512 per 100,000 live births) and adverse child outcomes.

**Policy gaps:** No integration of mental health screening into antenatal care protocols (WHO recommends universal screening using validated tools like the Edinburgh Postnatal Depression Scale), no trained midwives or community health workers in perinatal mental health, no mother–baby units in psychiatric facilities (forcing separation during hospitalisation) and no psychosocial interventions tailored to perinatal populations.

#### b. Internally displaced persons (IDPs)

Nigeria has 3.6 million IDPs due to Boko Haram insurgency, farmer–herder conflicts, banditry and other violence (IDMC, 2023). Conflict-exposed populations experience elevated rates of PTSD (25–40%), depression (20–35%) and anxiety disorders (15–30%) (Sheikh et al., 2015), yet mental health services in IDP camps and return communities are nearly absent.

**Policy Gaps:** Neither the 2013 policy nor the 2023 Act addresses humanitarian mental health or MHPSS in emergencies, no MHPSS in Emergencies strategy (unlike Uganda, which has operationalised IASC MHPSS guidelines), no psychological first aid training for humanitarian workers and no mobile mental health teams in conflict-affected states. IDPs in northeastern Nigeria face compounded trauma, with women and children disproportionately affected by gender-based violence and disrupted traditional support systems.

#### c. Rural and northern populations

Mental health service distribution favours southern, urban areas. Northern states – particularly Northwest and Northeast – have the lowest psychiatrist density (0.05 per 100,000 in some states versus 0.80 in Lagos), highest stigma levels and greatest cultural distance from biomedical mental health models (Abdulmalik et al., 2016).

Barriers include distance to facilities (average > 50 km to nearest psychiatric service in rural areas), linguistic barriers (services rarely available in Hausa, Kanuri, Fulfude or other northern languages), cultural incongruence (biomedical models conflict with Islamic and traditional explanatory frameworks) and economic barriers (transport costs prohibitive; opportunity costs of seeking care high).

**Policy Gaps:** No strategies for addressing geographic inequity beyond vague references to "expanding access," no cultural adaptation of interventions (most training materials assume a southern, Christian, English-speaking context) and no engagement with Islamic scholars or institutions (despite Islam's central role in northern Nigerian life).

#### d. Adolescents and young people

Youth (ages 10–24) comprise 30% of Nigeria's population and face elevated mental health risks: academic stress, unemployment, substance use (cannabis, tramadol and codeine), exposure to violence and social media pressures. Suicide rates among youth (15–29 years) are estimated at 15–20 per 100,000—higher than the general population (Okonkwo and Akpunne, 2023). UNICEF data indicate 1 in 6 young Nigerians (15–24 years) experience frequent depression, anxiety or worry (UNICEF x Gallup, 2021).

**Policy Gaps:** Policies mention "children" generically but lack adolescent-specific provisions (adolescence differs developmentally from childhood), no school mental health programmes (despite schools being ideal sites for early identification and intervention), no youth-friendly mental health services (existing services designed for adults; clinical environments alienate youth) and minimal attention to digital mental health platforms (which could reach tech-savvy youth).

#### e. Persons with severe mental illness and psychosocial disabilities

Individuals with schizophrenia, bipolar disorder, severe depression and other conditions face the most egregious rights violations: chaining in homes or spiritual healing centres, prolonged involuntary detention in psychiatric facilities without judicial review, forced treatment without informed consent and abandonment by families.

**Policy Gaps:** The 2023 Act strengthens legal protections but provides no resources for alternatives to institutionalisation, no community-based rehabilitation services (supported housing, vocational rehabilitation and intensive case management), no implementation mechanisms for the act's rights provisions

(no funding for independent mental health advocates, no judicial training on mental health law and no monitoring of rights violations) and no disability benefits or social protection for persons with psychosocial disabilities (unlike physical disabilities covered under Discrimination Against Persons with Disabilities Act 2018). Families often cannot afford ongoing treatment, lack knowledge about managing severe mental illness and face stigma. Traditional healing centres, though often well-intentioned, sometimes use harmful practices (chaining, beating and starvation). When families exhaust resources, abandonment occurs – leading to homelessness, exploitation or institutionalisation.

### Cross-cutting equity considerations

These analyses reveal that Nigerian mental health policy embodies a "one-size-fits-none" approach. Generic interventions fail to address specific vulnerabilities. Future policy must explicitly name vulnerable populations and detail targeted interventions (not generic statements like "services for all"), disaggregate monitoring data by gender, age, geography, conflict-exposure, IDP status and disability status (to identify inequities), involve affected populations in policy design through participatory processes (co-design, not consultation) and allocate resources proportionate to burden and need (not just population size – affirmative action for marginalised groups).

Without such differentiation, mental health policies risk entrenching rather than reducing inequities – as has occurred with Nigeria's maternal health policies, which disproportionately benefit educated, urban women while rural, poor women's mortality remains high (Onah et al., 2006).

## Conclusion and recommendations

### Summary of key findings

Nigeria's mental health policy has evolved from custodial and stigmatising frameworks (1916 Lunacy Ordinance) to rights-based, community-oriented approaches (2023 Mental Health Act). Policy content demonstrates alignment with international standards and recognition of mental health as a public health priority. However, persistent implementation deficits undermine these gains:

- **Context:** Fiscal constraints, federal–state coordination gaps, political instability and competing priorities limit operationalisation.
- **Content:** Policies lack cost plans, timelines, operational detail and context-specific adaptations.
- **Processes:** Top-down development excludes states, service users and traditional healers, reducing ownership and feasibility.
- **Actors:** Power asymmetries favour federal authorities and specialist professionals over states, communities and service users.

Consequently, workforce expansion is inadequate (0.16 psychiatrists per 100,000), treatment coverage is low (10–15%) and most Nigerians (85–90%) with mental disorders receive no professional care. Comparative analysis highlights the importance of dedicated authorities, community-based models and inclusive engagement.

### Evidence-informed SMART recommendations

Six SMART (Specific, Measurable, Achievable, Relevant, Timebound) recommendations addressing the identified implementation gaps:

### Recommendation 1: Increase Mental Health Budget Allocation

1. **Target:** Raise allocation from 3.3% to 5% of the total health budget by 2027, with phased increases: 4% (2025), 4.5% (2026) and 5% (2027).
2. **Mechanism:** Establish Federal Mental Health Budget Task Force; quarterly reporting; matching grants to incentivise states.
3. **Accountability:** Annual Mental Health Expenditure Report audited by the Office of the Auditor-General.

### Recommendation 2: Expand Mental Health Workforce Through Task-Shifting

1. **Target:** Train and deploy 1,000 mental health nurses and 500 community mental health officers by 2028.
2. **Mechanism:** Standardised WHO mhGAP-based curriculum, cascade training, mentorship and accreditation.
3. **Accountability:** Biannual progress reports published by the National Primary Healthcare Development Agency.

### Recommendation 3: Establish National Mental Health Information System (NMHIS)

1. **Target:** Operational NMHIS by 2026, integrated with NHMIS, monitoring service coverage, workforce, budget execution, treatment gaps and rights violations.
2. **Mechanism:** Pilot in six states, national rollout with public dashboards.
3. **Accountability:** Annual "State of Mental Health in Nigeria" report disseminated to stakeholders.

### Recommendation 4: Ensure State-Level Policy Adoption and Implementation

1. **Target:** All 36 states and FCT adopt aligned mental health policies by December 2026.
2. **Mechanism:** Federal technical and financial support, peer learning networks and template policies.
3. **Accountability:** The National Council on Health reviews progress quarterly; non-compliant states are publicly identified.

### Recommendation 5: Integrate Mental Health into Primary Healthcare

1. **Target:** PHC integration in 5,000 centres by 2029, prioritising underserved areas.
2. **Mechanism:** mhGAP training, essential medicines, supervision protocols and phased rollout.
3. **Accountability:** PHC readiness and service utilisation are tracked by the National Primary Healthcare Development Agency.

### Recommendation 6: Establish Inclusive Stakeholder Engagement Mechanisms

1. **Target:** National Mental Health Advisory Council by June 2026, minimum 30% representation from persons with lived experience.

2. **Mechanism:** Quarterly meetings, working groups and Secretariat support.
3. **Accountability:** Annual implementation and rights report submitted to the National Assembly and the President.

### Future research priorities

Critical gaps requiring evidence generation:

1. **Implementation Science:** Evaluate task-shifting, PHC integration and digital interventions using frameworks such as CFIR and RE-AIM.
2. **Health Economics:** Conduct cost-effectiveness analyses of community-based versus facility-based models; assess catastrophic health expenditures.
3. **Service User Perspectives:** Qualitative studies centring marginalised populations (rural women, IDPs and persons with severe mental illness).
4. **Traditional and Faith Healing:** Ethnographic research on practices, outcomes and potential for structured collaboration.
5. **Mental Health Information Systems:** Feasibility studies and development of culturally appropriate assessment tools.
6. **Workforce Retention:** Investigate retention factors and the effectiveness of retention strategies.

### Conclusion

Nigeria stands at a pivotal juncture: the 2023 Mental Health Act provides a strong legal and normative foundation, yet without political will, adequate resources, empowered implementation bodies and inclusive engagement, it risks remaining aspirational. Key imperatives include the following:

1. **Political prioritisation:** Elevate mental health above competing demands.
2. **Resource commitment:** Minimum 5% of the health budget and sustained financing.
3. **Structural reforms:** Establish an empowered Mental Health Commission and coordination mechanisms.
4. **Cultural humility:** Engage traditional and faith healing systems rather than marginalise them.
5. **Equity focus:** Prioritise women, IDPs, rural populations, youth and persons with severe mental illness.
6. **Accountability:** Robust monitoring, transparent reporting and consequences for non-implementation.

Implementation must go beyond legislation to cost plans, dedicated bodies, multi-stakeholder engagement, culturally appropriate adaptations, monitoring systems and sustained political and financial commitment. Mental health is essential for sustainable development, economic productivity, social cohesion and human dignity. Nigeria, with Africa's largest population and economy, has the opportunity to lead continental mental health system strengthening. The time for action is now.

**Open peer review.** To view the open peer review materials for this article, please visit http://doi.org/10.1017/gmh.2026.10158.

**Data availability statement.** All data analysed in this review are publicly available from the cited sources. A complete list of included documents is provided in the reference list.

**Author contribution.** Benmun Damul, Hazel King and Halima Jafiya contributed equally to the conception, design and writing of this scoping review. All authors read and approved the final manuscript.

**Financial support.** This research received no external funding.

**Competing interests.** The authors declare no conflicts of interest.

**Ethics statement.** This scoping review analysed publicly available documents and published literature and did not involve human subjects. Therefore, ethical approval was not required.

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
