## [Reviewer Report]

Review of Manuscript GMH-2025-0219: “Mental Health Policy Interventions in Nigeria and Their Impact on Mental Health Service Delivery”

General Comments

This manuscript provides a broad and ambitious overview of the mental health policy landscape in Nigeria, charting a course from the colonial-era Lunacy Ordinance to the landmark 2023 Mental Health Act. The topic is undeniably timely and significant, and the authors have compiled a wide range of material to construct their narrative.

The paper’s strengths lie in its historical scope and its correct identification of the key, persistent challenges facing the Nigerian mental health system: chronic underfunding, severe workforce shortages, and pervasive socio-cultural stigma.

However, despite these merits, the manuscript in its current form falls substantially short of the analytical rigour and methodological transparency required for publication in a leading journal like Cambridge Prisms: Global Mental Health. The work functions primarily as a descriptive, narrative summary of existing information rather than a critical, analytical review that advances scholarly understanding. It recycles known challenges and policy milestones without offering novel insights, a guiding theoretical framework, or a robust evidentiary synthesis.

MAJOR CONCERNS

1. Absence of Methodological Rigour

The manuscript is presented as an “Overview Review,” yet it entirely lacks a methods section. This is a foundational flaw. Without methodological transparency, the paper has limited credibility, reproducibility, and scholarly value.

• It is necessary for the authors to ass a dedicated “Methods” section. This section should clearly: (a) Define the review type (e.g., narrative review, scoping review, critical policy analysis) and justify your choice; (b) Detail the search strategy, including databases searched, the time window, and key search terms; and (c) Specify the inclusion and exclusion criteria for selecting sources and explain your approach to handling grey literature and non-peer-reviewed materials.

2. Lack of Analytical Depth and Theoretical Framing

This is the most pervasive weakness of the manuscript. The review describes what happened but rarely interrogates why certain policies failed while others succeeded, how power and institutions shaped outcomes, or which mechanisms link policy to delivery.

• I will advice that the authors anchor the entire manuscript in a recognized analytical framework. For example: (a) Use Walt and Gilson’s Policy Triangle to systematically analyse the content, context, process, and actors involved in Nigeria’s policy reforms; (b) Apply an implementation science framework (e.g., CFIR) to move beyond merely listing barriers and instead analyse the complex facilitators and impediments to implementing policies like PHC integration; and (c) Engage with postcolonial and decolonial theory to offer a more critical analysis of how the colonial legacy of the Lunacy Act perpetuates custodial logics in the present day, a point you touch upon but do not develop.

3. A Flawed and Inconsistent Evidence Base

The manuscript is undermined by numerous issues related to its use of data and sources, which compromise its reliability. Examples are:

a. Statistical Inconsistencies: The paper cites multiple conflicting figures for the number of practising psychiatrists (alternately “about 200,” “only about 250,” and “approximately 350 by 2020”) without reconciliation;

b. Imprecise and Vague Claims: The headline statistic that mental disorders “affect at least approximately 20% of the population” is methodologically vague, awkwardly phrased, and repeated without proper sourcing or explanation of the metric (e.g., lifetime vs. 12-month prevalence);

c. Use of Outdated Data: The reliance on the 2006 WHO-AIMS report for facility and bed statistics is unacceptable without explicitly flagging that the data is nearly two decades old and discussing the implications of this data gap; (

d. Over-reliance on Non-Scholarly Sources: Major claims are often supported by newspaper articles, online think-pieces, and aggregator sites, which are not appropriate for anchoring the central arguments of a scholarly review.

• For these, it is best if the authors conduct a thorough audit of all data presented. They should harmonise all statistics, use a single time-stamped best estimate for each indicator, and transparently cite the source. They should also prioritise peer-reviewed sources and authoritative reports for all central claims.

4. Failure to Demonstrate “Impact”

The title promises an analysis of “impact,” but the manuscript delivers only correlation. You describe policy changes and then describe incremental improvements in the system, but you do not build a convincing case that the former caused the latter.

• The impact section should be reframed to build a plausible causal narrative. For each policy, specify the intended instruments and mechanisms, and then critically assess the available evidence for intermediate and long-term outcomes. Where causal inference is not possible, this must be explicitly stated. A logic model figure would be highly effective here.

MODERATE CONCERNS

5. Structural Flaws, Repetitiveness, and Excessive Length

The manuscript is overlong, unfocused, and highly repetitive. There is significant thematic overlap between the “Current State of Mental Health Services” and “Barriers to Implementation” sections, weakening the narrative thrust. The paper also lacks key structural elements, such as a distinct Discussion section to separate findings from interpretation.

• I suggest the authors undertake a substantial restructuring and editing process. The text should be condensed by at least e the text by at least 30% by eliminating repetitive statements about funding, stigma, and workforce deficits.

• They should also restructure the paper to follow a clear Introduction → Methods → Findings → Discussion → Conclusion format.

• They should merge and streamline the overlapping content in the “Current State” and “Barriers” sections

6. Generic and Unactionable Recommendations

The recommendations section is one of the weakest parts of the paper, offering generic advice that could apply to any LMIC (e.g., “increase funding,” “strengthen legislation”) and adds little value.

• There is need to replace the current list with SMART (Specific, Measurable, Achievable, Relevant, Time-bound) recommendations. Ground each one in the evidence you have presented.

7. Superficial Comparative Analysis and Global Context

While the inclusion of a comparative section is a strength, its execution is thin and it appears late in the paper as an add-on rather than an integrated analytical lens. The paper fails to adequately situate Nigeria’s experience within global research and learnings, as required by the journal.

• I suggest to deepen the comparative analysis by focusing on 2-3 well-matched countries and systematically comparing policy design, financing mechanisms, and outcomes. The authors should weave these comparative insights throughout the manuscript, not just in one section. In the Discussion, they should explicitly articulate what the global mental health community can learn from Nigeria’s specific successes and failures.

8. Lack of Nuance and Overgeneralisation

The manuscript often treats “Nigeria” and its “culture” as monolithic entities, failing to engage with the nation’s immense diversity. It also does not adequately address the intersectional vulnerabilities of specific subpopulations (e.g., women, IDPs, rural communities).

• The authors need to integrate a more nuanced analysis of subpopulation differences. When discussing stigma and traditional beliefs, they should move beyond generalisations and engage with the rich anthropological and sociological literature on the topic in specific Nigerian contexts.

Conclusion

The manuscript addresses a topic of vital importance. However, to be publishable, it requires a fundamental transformation from a descriptive summary into a rigorous, methodologically transparent, and critically analytical review.

The authors must define their methodology, adopt an analytical framework, overhaul the evidence base, and sharpen their recommendations.

---

## [Reviewer Report]

Damul et al provide an articulate review of mental healthy system challenges in Nigeria. The prose is well-written and engaging. The piece is somewhat repetitive and would be improved if shortened (two specific examples: 1. first sentence of Introduction vs. second sentence of first paragraph of “The Scope of Mental Health Issues in Nigeria”; 2. paragraph beginning “Cultural factors present...” echoes discussion of cultural beliefs two pages earlier; there are many other examples of repeating discussions of under funding, personnel shortages, etc). Please better define “task-shifting”. This is a complete and timely review and I strongly support its publication.

---

## [Reviewer Report]

Thank you for your very important paper on mental health policy and service delivery in Nigeria. While this is a very timely manuscript, it is quite long and can be shortened in order to keep you reader engaged.

---

## [Editor Report]

Dear Authors 

We have received reviewer comments now, and recommend major revision to your manuscript in order for us to consider it for publication.

Regards

Siham

---

## [Reviewer Report]

The authors have carried out requested revisions in a substantial manner and i believe the revised manuscript should be accepted for publication

---

## [Reviewer Report]

This draft has been substantially revised, along the lines suggested by the first reviewer for a systematic review: I will defer to that reviewer’s expertise in assessing whether the paper meets all current standards for such reviews. The draft is improved in all regards, including removal of repeated content, and it now has a clear and effective organization. The paper is still quite long but its thoroughness and timeliness justify the length in my mind (I would rather one integrated paper than several separate ones). I strongly urge publication.

---

## [Reviewer Report]

Thank you for the significant improvement in the manuscript. my major areas of concerns concerning it being engaging and being shortened has been addressed.

---

## [Editor Report]

Dear Authors 

Thanks for your revised manuscript and addressing reviewers comments. Pleased to inform you that we accept it for publication; will follow up with next steps. 

Warmly 

Siham